# Detection of *Xylella fastidiosa* in Host Plants and Insect Vectors by Droplet Digital PCR

Serafina Serena Amoia [1,2], Angelantonio Minafra [1], Angela Ligorio [1], Vincenzo Cavalieri [1], Donato Boscia [1], Maria Saponari [1] and Giuliana Loconsole [1,*]

1   Institute for Sustainable Plant Protection (IPSP)—National Research Council, 70126 Bari, Italy
2   Department of Soil, Plant and Food Sciences, University of Bari Aldo Moro, 70126 Bari, Italy
*   Correspondence: giuliana.loconsole@ipsp.cnr.it

**Abstract:** *Xylella fastidiosa* (*Xf*) is a Gram-negative plant bacterium that causes severe diseases affecting several economically important crops in many countries. To achieve early detection of the pathogen, a droplet digital PCR (ddPCR)-based approach was used to detect the bacterium at low concentrations in different plant species and insect vectors. In this study, we implemented the reaction conditions of a previously developed ddPCR assay, and we validated its use to detect *Xf* in insect vectors as well as in a broader list of host species. More specifically, the sensitivity and accuracy of the protocol were assessed by testing five plant matrices (*Olea europaea*, *Nerium oleander*, *Vitis vinifera*, *Citrus sinensis*, and *Prunus dulcis*), and for the first time, the insect vector (*Philaenus spumarius*), was either naturally infected or artificially spiked with bacterial suspension at known concentrations. The lowest concentrations detected by ddPCR were 5 ag/µL of bacterial DNA and $1.00 \times 10^2$ CFU/mL of bacterial cells. Both techniques showed a high degree of linearity, with $R^2$ values ranging from 0.9905 to 0.9995 and from 0.9726 to 0.9977, respectively, for qPCR and ddPCR. Under our conditions, ddPCR showed greater analytical sensitivity than qPCR for *O. europea*, *C. sinensis*, and *N. oleander*. Overall, the results demonstrated that the validated ddPCR assay enables the absolute quantification of *Xf* target sequences with high accuracy compared with the qPCR assay, and can support experimental research programs and the official controls, particularly when doubtful or inconclusive results are recorded by qPCR.

**Keywords:** droplet digital PCR; ddPCR; qPCR; Xylella fastidiosa; quarantine pest; molecular diagnosis





## 1. Introduction

*Xylella fastidiosa* (*Xf*) is a Gram-negative bacterium belonging to the *Xanthomonadaceae* family [1] which colonizes the xylem vessel of plants and the foregut of insect vectors. This pathogen has a very wide host range, including over 600 species [2], and is well known as a causal agent of detrimental crop diseases, such as Pierce's disease in grapevine (PD), citrus variegated chlorosis (CVC), almond leaf scorch (ALS), and olive quick decline syndrome (OQDS), a recently described disease decimating olives in southern Italy [3], Brazil [4], Argentina [5], and the Balearic Islands [6]. In 2013, the introduction of *X. fastidiosa* subsp. *pauca* (*Xfp*) ST 53 in Salento Peninsula (Apulia, Italy) represented the first confirmed outdoor outbreak of this exotic bacterium in Europe, leading to a dramatic epidemic of OQDS [3,7]. Long-distance dispersal of *Xf* occurs mainly via human-mediated movement of infected plants, propagating material and infected vectors (i.e., hitchhiking in cars or trucks), whereas plant-to-plant transmission over short distances relies on xylem sap-feeding insect vectors, with *Philaenus spumarius* [8] being the predominant species in the European and Mediterranean countries. In the Apulia region, the quick spread of the pathogen has been dramatically eased by abundant vector populations, the occurrence of extensive monocultures of two autochthonous susceptible olive cultivars (Cellina di Nardò and Ogliarola salentina), and favorable climatic conditions [9,10]. The detrimental



impacts of bacterial infection entail that, at the European level and in several other countries worldwide, the bacterium is categorized as a quarantine and priority pest, necessitating the consequent adoption of mandatory preventive and containment measures. Preventive strategies rely on the early detection of infected plants followed by the prompt removal of infected sources, reducing risks for further spread and for the establishment of new foci. Currently, several diagnostic approaches are available to detect the pathogen in host plants, including isolation and culturing on artificial media, the use of polyclonal antisera in serological tests, and different molecular techniques (i.e., conventional and quantitative PCR, LAMP, and RPA) [11–14]. Molecular tests are the most commonly used, especially in European countries where only molecular tests are recognized as official diagnostic tests (Reg. 2020/1201—Annex IV). Indeed, the detection of the bacterium in insect vectors is mainly based solely on molecular methods. Among the official molecular tests available for *Xf*, quantitative PCR (qPCR) is widely used, and the assay based on the protocol developed by Harper et al. (2010) [11] is considered the most sensitive regardless of the host plant species EPPO PM 7/24 (4); [15]) and the bacterial subspecies. However, the interpretation of the qPCR results (quantitation cycle—Cq) in samples with low bacterial concentration and the estimations of the bacterial population (absolute quantification) in the positive samples remain two challenging aspects of the qPCR tests.

Droplet digital polymerase chain reaction (ddPCR) is an advanced and powerful molecular technology that allows for accurate detection and absolute quantification of the nucleic acid present in samples, even when the target is present at a very low level. The principle of digital PCR was first described in the 1990s [16,17] and, later on, was further optimized for different diagnostic purposes. Differently from the qPCR, ddPCR does not require a calibration curve nor the use of housekeeping genes to normalize the results, thus reducing the reaction requirements and facilitating the data analysis and comparability [18,19]. The ddPCR technology massively distributes the sample into thousands of independent nanoliter-sized droplets in a random approach. Each droplet acts as an individual PCR test tube in which amplification can take place [20,21]. Partitions containing amplified fluorescent products are considered positive, whereas those without fluorescence are considered negative. The absolute number of target DNA molecules in a sample can be calculated directly from the ratio of positive droplets to total partitions using Poisson's law of small numbers [18,19]. Although this molecular technique is more expensive and time-consuming than qPCR, the high accuracy and sensitivity support its application in the diagnosis of microorganisms in the early stages of infection. Indeed, timely detection of new outbreaks of quarantine pests is fundamental to avoid the establishment of exotic pathogens in new areas. Digital droplet PCR has been shown to achieve more precise detection results than qPCR, and this molecular method has now gained interest in different biological applications for research and diagnostic purposes. Our work aimed to implement the use of ddPCR for the detection and quantification of *Xf* in different susceptible plant species, especially in insect vectors, as the latter are very rich in PCR inhibitors and contaminants. The sensitivity and efficiency were compared to those obtained by qPCR using the same DNA preparations.

## 2. Materials and Methods

### 2.1. Bacterial Strain

The reference bacterial strain 'De Donno', belonging to the subsp. pauca (Xfp) and harboring the sequence type (ST) 53, was isolated in the Apulia region (Italy) in 2016 from an olive tree affected by olive quick decline syndrome (OQDS) [22]. It was used as target strain to prepare artificially contaminated plant sap containing known concentrations of bacterial cells/copies of the target DNA. Bacterial colonies, grown on PD3 solid medium at 28 °C for 7 days [23], were scraped from plates and dispersed in sterile demineralized water to prepare a bacterial suspension with an optical density (OD) at 600 nm of 0.5 OD. This suspension was diluted 1:4 (approx. 108 CFU/mL) and used to contaminate plant saps recovered from all the matrices selected in this study. The concentration of the bacterial suspension (CFU/mL) was also determined by plate counting.

## 2.2. Plant Sources and Insects

Plant matrices used toevaluate ddPCR assay were collected from *Xf*-free sources of *Olea europea* L., *Vitis vinifera* L., *Citrus sinensis* (L.) Osbeck, *Prunus dulcis* Mill., and *Nerium oleander* L., grown in the greenhouse. For the insect vectors, adult specimens of *Philaenus spumarius* L. (*Hemiptera*: *Aphrophoridae*) were collected in fields located in an *Xf*-free area of the Apulia region. The accuracy of ddPCR assays was also evaluated on naturally infected olive and insect samples collected in the framework of the regional monitoring program carried out in Apulia in 2021–2022. The panel of naturally infected olive samples was selected in order to obtain a full representation of samples with high, medium, and low bacterial concentrations.

## 2.3. Sample Preparation and DNA Extraction

Plant tissues used in our tests consisted of leaf tissues (either petioles, midribs, or basal leaf parts) for oleander, grapevine, and citrus; stem/twig portions of 1–1.5 cm long for olive; and scraped xylem tissue from hardwood cutting for almond. Insect samples were prepared by removing the head from the body (EPPO 7/24 (4) [15]). Plant and insect tissues were homogenized in extraction buffer (CTAB buffer) following the standard ratio (*w:v*) [15]. Artificially contaminated samples were prepared by spiking both the plant sap and the macerated insect with the *Xf* suspension previously described in order to obtain 3 replicates/matrices of 10-fold serial dilutions containing from $10^7$ to 10 CFU/mL. Both the spiked and the naturally infected samples were processed using the Maxwell® RSC PureFood GMO and Authentication Kit (Promega), following the manufacturer's instructions to extract DNA from plants and insects. For insects, individual excised heads were homogenized in 500 µL of CTAB buffer [15], and DNA was purified using the same kit described for plant tissues, with the only difference being that the total DNA was eluted in 50 µL of elution buffer (EB, Promega). Negative and positive internal controls (NIC, PIC) for each matrix were prepared and processed as described previously. The PIC was contaminated with a bacterial suspension $10^7$ CFU/mL before the DNA extraction.

## 2.4. Quantitative PCR Assay

For qPCR assays, the procedure described in Annex 5—EPPO PM 7/24 (4) [15]—was based on the set of primers/probe designed by Harper et al. (2010) [11], but was slightly modified. Briefly, the reaction mix was prepared in a final volume of 11 µL containing: 1x TaqMan™ Fast Advanced Master Mix (Applied biosystem), 300 nM of each primer (XF-F/XF-R), 200 nM of 6-FAM/BHQ-1 labeled probe (XF-P), and 1 µL of DNA extract. Positive and negative amplification controls (i.e., PAC, NAC) were included in each test. The amplification conditions were 95 °C for 5 min, followed by 40 cycles of 94 °C for 10 s and 62 °C for 40 s. Tests were performed on the thermal cycler CFX96 Touch Real-Time PCR Detection System (Bio-rad, Hercules, CA, USA) using 96-well plates (Bio-rad). For artificially contaminated samples, each sample/replicate was amplified in triplicate and each replicate/matrix (including NIC) was tested in at least three independent assays. DNA recovered from the naturally infected samples was tested in duplicate wells. Data acquisition and analysis were performed using the CFX Maestro 1.1 version N. 4.1 software (Bio-rad, Hercules, CA, USA ). The qPCR amplification efficiency was estimated for each matrix from the slopes of the standard curves generated by the 10-fold serial dilutions, using the equation $E = 10-1/\text{slope} \pm 1$. The limit of quantification (LOQ) of the assay was defined based on the lowest dilution yielding positive results for the three replicates tested when these were in the linear range of the standard calibration curve (i.e., when the $\Delta Cq$, among the dilution considered and the previous one tested, was close to 3). The limit of detection (LOD) corresponded to the lowest serial dilution yielding a positive result in more than 50% of the overall replicates of each matrix tested in all independent assays. A recombinant plasmid containing a region of the *rimM* gene targeted by the primers/probe reported by Harper et al. (2010) [11], was used to set up PCR reactions

with known concentrations of target DNA (ng/μL) and, from there, to calculate the copy number of *Xf* bacterial DNA.

### 2.5. Droplet Digital PCR Assay

Digital droplet PCR was performed on QX200 Droplet Digital PCR System (Bio-Rad) according to the manufacturer's instructions [21], using the primers/probes of Harper et al. (2010) [11]. To implement the performance of the ddPCR assay reported by Dupas et al. (2019) [24], i.e., to achieve a better separation of positive and negative droplet clusters without 'droplet rain' in naturally infected olive plant and insect samples, different volumes of plant/insect extracts (2 μL, 4 μL, 6 μL, and 8 μL ) were tested using ddPCR conditions reported by Dupas et al., 2019. Once the optimal amount of purified DNA of plants and insects was determined, 3 different primers/probe concentrations (900 nM/300 nM; 600 nM/300 nM; and 300 nM/300 nM) were tested in 20 μL of the reaction mix (2x ddPCR™ Supermix for Probes (No dUTP)) using both spiked and naturally infected olive and insect samples as templates. The reaction mix (20 μL) and 70 μL of droplet-generating oil (Bio-rad) were added to a droplet-generating DG8 cartridge and loaded onto Bio-rad Automated Droplet Generator (Bio-rad). The water-in-oil droplets (40 μL) were carefully transferred to a 96-well PCR plate, which had been heat-sealed at 175 °C with a pierceable foil using a PX1™ PCR Plate Sealer (Bio-rad) and then placed in a C1000 Thermal Cycler (Bio-rad) for end-point PCR. Amplification reactions were performed with the cycling parameters optimized by Dupas et al. (2019) [24], except for increasing the number of cycles to 45 with a temperature ramp rate of 2 °C/s. After amplification, the PCR plate was directly transferred to the droplet reader (QX200TMDroplet DigitalTM System (Bio-rad)) set in the absolute quantification (ABS) modality. When the assay was carried out on spiked matrices, each replicate/matrix of the 10-fold serial dilution from $10^6$ to 10 CFU/mL was amplified in duplicate in 3 different runs. Then, for each replicate, samples with concentrations close to the LOQ, identified by qPCR, were amplified in duplicate in several separated runs to evaluate the reproducibility of the results between different amplification reactions. The LOD was calculated as previously described. Finally, field plant and insect samples were checked by ddPCR in duplicate in a single run.

### 2.6. Evaluation of ddPCR Performance in Naturally Infected Plant and Insect Samples

The performance efficiency of both ddPCR and qPCR assays to detect *Xf* in naturally infected samples from olive trees and insects was assessed using 34 olive samples from the field and 27 specimens of *P. spumarius* collected by sweep net in the demarcated infected area of Apulia in July 2022. DNA from plants and insects was extracted as previously described. Each sample was then amplified in duplicate in the same amplification run. In addition, negative controls (samples collected in an *Xf*-free area) were included in each test to evaluate the presence of PCR contaminants.

### 2.7. Data Analysis

The linear regression of the standard curves generated by qPCR assay was calculated by CFX™ Maestro Software version 2.2 (Bio-Rad). For ddPCR, fluorescent signals of droplets were acquired, expressed in target copies/μL of the reaction, and analyzed by the QX-Manager-V.1.2-STD software (Bio-Rad). For each ddPCR experiment, positive droplets, with higher fluorescent signals, and negative droplets, with lower fluorescent signals (considered as background), were divided by applying an amplitude threshold line; ddPCR reactions with fewer than 10.000 generated droplets were excluded from the analysis, and a reaction was considered positive if at least three positive droplets were counted [25]. Poisson's statistic was used to calculate the absolute concentration in each sample as a copy number/μL of the target DNA. The error reported for every single well was represented as the Poisson assessed at a 95% confidence level. The linear range of the ddPCR assay was determined by plotting the number of expected target copies/μL vs. the number of target copies/μL generated by ddPCR using both serial dilutions of the plasmid

DNA and the bacterial suspensions. Pearson's correlations and linear regression were also used to evaluate the relationship between the results of ddPCR and qPCR assays.

## 3. Results

### 3.1. Optimization of the ddPCR Assay

Preliminary experiments included the use of different amounts of DNA preparations from plants and insects (Supplementary Figure S1). Once the optimal number of samples was defined, different primers/probe concentrations were tested in an attempt to improve the detection of the bacterium in spiked and naturally infected olive and insect samples. By increasing the number of cycles to 45, with respect to the study reported by Dupas et al. (2019) [24], the optimal primer/probe concentrations and DNA quantities were found to correspond to 600/300 nM (primers/probe) and 4 and 6 µL of purified plant/insect DNA, respectively. These reaction conditions proved to be suitable for detecting *Xf* in all six matrices analyzed, with an optimal difference in the fluorescence signals between positive and negative droplets and generating a high number of amplification products (number of positive droplets) (Figure 1A–F). When compared to qPCR, ddPCR yielded a narrower linearity range, from $10^6$ to $10^1$ CFU/mL, since the high concentration of positive droplets saturated the fluorescence signal at concentrations above $10^6$ CFU/mL, making the Poisson algorithm invalid. Negative and positive controls correctly produced the expected results, with no fluorescence signal detected in NIC or NAC.

### 3.2. Linearity and Analytical Sensitivity of qPCR Assay

Ten-fold serial dilutions of both plasmid DNA and *Xfp* bacterial suspension were used to generate the standard curves for the qPCR tests (Figure 2A,B). When testing artificially contaminated samples, containing either plasmid DNA (50 pg/µL–50 ag/µL) or bacterial suspension ($1.00 \times 10^7$–$1.00 \times 10^1$), the recovered curves showed good linearity, with $R^2$ values of 0.9943 and 0.9973 for the *rimM*-recombinant DNA plasmid and the bacterial suspension, respectively. The qPCR efficiency values retrieved from these standard curves were 100.72% and 116.96% for plasmid DNA and bacterial suspension, respectively. According to the standard curves, the analytical sensitivity of the qPCR test was 50 ag/µL, corresponding to 9.45 copies/µL, and $1.00 \times 10^2$ CFU/mL, corresponding to 0.1 copies/µL, for plasmid DNA and bacterial cells, respectively.

### 3.3. Linearity and Analytical Sensitivity of the ddPCR Assay

Ten-fold serial dilutions of *rimM*-recombinant plasmid DNA and *Xfp* bacterial suspension were used to construct the regression curves for the ddPCR assay. As shown in Figure 3A,B, the trend line of copies/µL obtained by ddPCR compared to our expectation indicated a high level of linearity in the evaluated range of concentrations, with an $R^2$ value of 0.9981 for the bacterial suspension and 0.9928 for the plasmid DNA. The detection limits of the assay were determined to be $4.30 \times 10^{-1}$ copies/µL and $5.06 \times 10^{-1}$ copies/µL for the bacterial suspension and the plasmid, respectively, corresponding to $8.60 \times 10^0$ and $1.01 \times 10^1$ copies per reaction in a 20 µL volume. The correlation values between the expected and measured concentrations were significant (*p*-value < 0.00001 at *p* < 0.05), with r = 1 for both bacterial suspension and *rimM* plasmid DNA dilutions.

### 3.4. Comparison between ddPCR and qPCR Assays in Artificially Contaminated Samples

The quantitative PCR and ddPCR assays carried out under the conditions implemented in this work exhibited optimal performance values, with high determination coefficients (Tables 1 and 2). Based on the standard and regression curves generated by the qPCR and ddPCR assays (Supplementary Figures S2 and S3), the linearity, LOQ, and LOD of both assays were determined. As reported in Tables 1 and 2, the qPCR and ddPCR assays exhibited good linearity for all the analyzed matrices, with ($R^2$) ranging from 0.9905 (for almond) to 0.9995 (for insects) and from 0.9743 (for insect) to 0.9985 (for almond), respectively. The slope value of the standard curve generated by qPCR for each matrix ranged from

−2.94 for oleander to −3.1993 for olive, corresponding to efficiency values of 118.84% and 105.38%, respectively. *Xfp* was correctly detected by qPCR in all three replicates/matrix, containing the lowest bacterial concentration, with LOQ and LOD being equal. The LOQ and LOD values recorded for ddPCR were identical for all plant matrices, while they differed in the case of the ddPCR assays on insects. More specifically, for the insects, LOQ corresponded to 10 CFU/mL (i.e., $9.22 \times 10^{-1}$ copies/μL), but at this concentration, the bacterium was detected in 4 out of 8 replicates. As such, $10^2$ CFU/mL (corresponding to 2.02 copies/μL) was considered to be the LOD, given that at this concentration, all 8 replicates tested positive. As shown in Tables 3 and 4, the ddPCR assay showed generally higher analytical sensitivity compared to the qPCR test in the range of one order of magnitude, detecting *Xfp* in the replicates containing lower bacterial concentrations than qPCR for olive (up to 10 CFU/mL, corresponding to 1.2 copies/μL in 4 of 7 replicates), citrus (up to $10^2$ CFU/mL corresponding to 0.24 copies/μL in 9 of 10 matrices), and oleander (up to $10^2$ CFU/mL corresponding to 0.922 copies/μL in 7 of 8 replicates). Equivalent LOQ and LOD levels ($10^2$ CFU/mL) were recorded in *V. vinifera* and *P. dulcis*, corresponding in ddPCR to $4.74 \times 10^{-1}$ and $3.92 \times 10^{-1}$ copies/μL, respectively. Indeed, *Xfp* was correctly detected in all replicates containing $10^2$ CFU/mL in *V. vinifera* by qPCR (3/3) and ddPCR (12/12). However, in *P. dulcis*, the bacterium was detected by ddPCR in 6 of the 10 replicates containing $10^2$ CFU/mL, and by qPCR in all 3 replicates containing $10^2$ CFU/mL. No positive droplets and no amplification curves were produced in the NTC or NIC for any of the matrices considered. The accuracy of the ddPCR assay when testing the spiked matrices reached 100%.

**Table 1.** Equation of regression curves and $R^2$ values generated by ddPCR for the six matrices spiked with *Xfp* bacterial suspension.

| ddPCR | Curve Equation | $R^2$ |
|---|---|---|
| *Olea europea* | $y = 1635.4 \times 10^{-1.278x}$ | $R^2 = 0.9867$ |
| *Vitis vinifera* | $y = 6607.7 \times 10^{-1.922x}$ | $R^2 = 0.9885$ |
| *Citrus sinensis* | $y = 2220.3 \times 10^{-1.86x}$ | $R^2 = 0.9949$ |
| *Prunus dulcis* | $y = 4432.9 \times 10^{-1.854x}$ | $R^2 = 0.9985$ |
| *Nerium oleander* | $y = 653.8 \times 10^{-1.392x}$ | $R^2 = 0.9949$ |
| *Philaenus spumarius* | $y = 25,107 \times 10^{-1.814x}$ | $R^2 = 0.9743$ |

**Table 2.** Equation of standard curves, $R^2$ value, and efficiency generated by qPCR for the six matrices spiked with *Xfp* bacterial suspension.

| qPCR | Curve Equation | $R^2$ | Efficiency |
|---|---|---|---|
| *Olea europea* | $y = -3.1993x + 38.963$ | 0.9962 | 105.38% |
| *Vitis vinifera* | $y = -3.1095x + 38.341$ | 0.9987 | 109.70% |
| *Citrus sinensis* | $y = -3.1376x + 38.847$ | 0.9993 | 108.31% |
| *Prunus dulcis* | $y = -3.0538x + 38.553$ | 0.9905 | 112.55% |
| *Nerium oleander* | $y = -2.94x + 37.563$ | 0.9933 | 118.84% |
| *Philaenus spumarius* | $y = -3.0427x + 38.005$ | 0.9995 | 113.14% |

**Table 3.** Mean concentrations were estimated in copies/μL, as measured by ddPCR in each serial dilution of the spiked plant and insect matrices.

| Dilution Range | *Olea europea* | | *Vitis vinifera* | | *Citrus sinensis* | | *Prunus dulcis* | | *Nerium oleander* | | *Philaenus spumarius* | |
| CFU/mL | *copies/μL* | *Replicates** | *copies/μL* | *Replicates** | *copies/μL* | *Replicates** | *copies/μL* | *Replicates** | *copies/μL* | *Replicates** | *copies/μL* | *Replicates** |
|---|---|---|---|---|---|---|---|---|---|---|---|---|
| $1.00 \times 10^6$ | $6.09 \times 10^2$ | 2/2 | $7.97 \times 10^2$ | 2/2 | $3.74 \times 10^2$ | 2/2 | $6.62 \times 10^2$ | 2/2 | $2.21 \times 10^2$ | 2/2 | $4.99 \times 10^3$ | 2/2 |
| $1.00 \times 10^5$ | $1.08 \times 10^2$ | 5/5 | $2.00 \times 10^2$ | 2/2 | $5.43 \times 10^1$ | 2/2 | $1.28 \times 10^2$ | 2/2 | $3.90 \times 10^1$ | 2/2 | $7.73 \times 10^2$ | 2/2 |
| $1.00 \times 10^4$ | $4.36 \times 10^1$ | 5/5 | $1.98 \times 10^1$ | 2/2 | $7.79 \times 10^0$ | 2/2 | $1.33 \times 10^1$ | 2/2 | $6.37 \times 10^0$ | 2/2 | $1.20 \times 10^2$ | 2/2 |
| $1.00 \times 10^3$ | $5.10 \times 10^0$ | 11/11 | $2.55 \times 10^0$ | 12/12 | $1.08 \times 10^0$ | 10/10 | $3.24 \times 10^0$ | 10/10 | $2.02 \times 10^0$ | 8/8 | $8.33 \times 10^0$ | 8/8 |
| $1.00 \times 10^2$ | $2.40 \times 10^0$ | 14/14 | $4.74 \times 10^{-1}$ | 12/12 | $2.43 \times 10^{-1}$ | 9/10 | $3.92 \times 10^{-1}$ | 6/10 | $9.22 \times 10^{-1}$ | 7/8 | $2.02 \times 10^0$ | 8/8 |
| $1.00 \times 10^1$ | $1.20 \times 10^0$ | 4/7 | n.d. ** | | n.d. ** | | n.d. ** | | n.d. ** | | $9.22 \times 10^{-1}$ | 4/8 |

* Replicates: number of positive replicates/number of replicates analyzed. ** n.d.: not detectable.

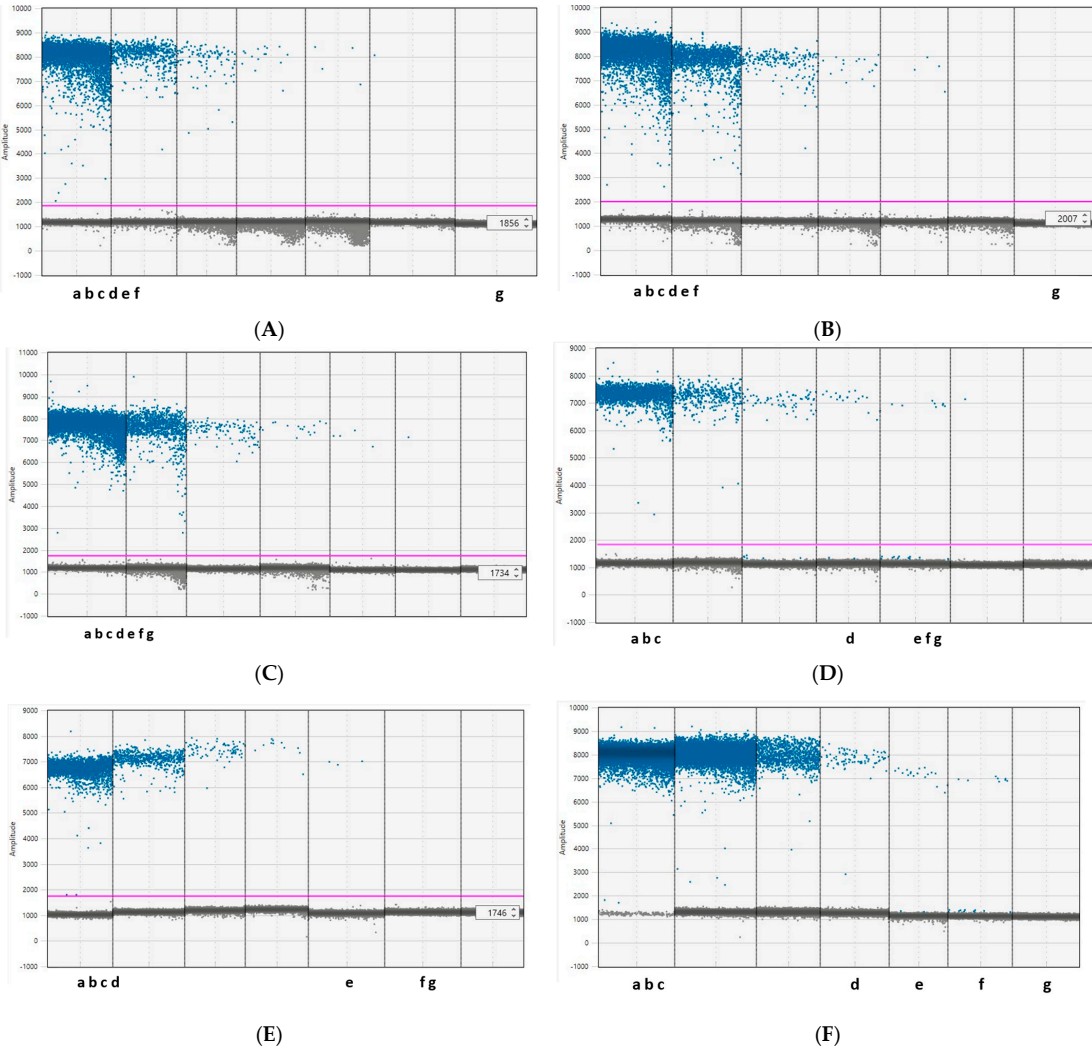

**Figure 1.** Comparison of the different limits of detection of *Xf* obtained by ddPCR in the five plant spiked matrices and in the insect vector at the optimized ddPCR parameters (600/300 nM primer/probe concentration and 45 end-point PCR cycles). Blue dots represent positive droplets above the pink horizontal threshold line. Gray dots represent the negative droplet background, with no amplification. (**A**): *O. europaea*; (**B**): *V. vinifera*; (**C**): *P. dulcis*; (**D**): *N. oleander*; (**E**): *C. sinensis*; and (**F**): *P. spumarius*. On *x-axis*: ten-fold dilution of *Xf* suspension reported in wells a to f; **a**: $10^6$ CFU/mL; **b**: $10^5$ CFU/mL; **c**: $10^4$ CFU/mL; **d**: $10^3$ CFU/mL; **e**: $10^2$ CFU/mL; and **f**: 10 CFU/mL. Well **g**: NIC (negative internal control), specific for each matrix; on the *y-axis*: amplitude value.

**Table 4.** Mean Cq values of the three replicates of 10-fold serial dilutions (CFU/mL) produced by qPCR Harper et al. (2010), modified for the spiked plant and insect matrices.

| Dilution Range CFU/mL | *Olea europea* | *Vitis vinifera* | *Citrus sinensis* | *Prunus dulcis* | *Nerium oleander* | *Philaenus spumarius* |
|---|---|---|---|---|---|---|
| $1.00 \times 10^6$ | 22.67 | 22.97 | 26.20 | 23.02 | 26.02 | 22.92 |
| $1.00 \times 10^5$ | 26.24 | 25.61 | 29.53 | 26.19 | 28.60 | 25.69 |
| $1.00 \times 10^4$ | 29.74 | 29.04 | 32.67 | 30.23 | 31.31 | 28.84 |
| $1.00 \times 10^3$ | 32.76 | 32.23 | 35.61 | 32.25 | 34.92 | 31.90 |
| $1.00 \times 10^2$ | 35.41 | 35.21 | n.d | 35.26 | n.d | 35.03 |
| $1.00 \times 10^1$ | n.d. | n.d | n.d | n.d | n.d | n.d |

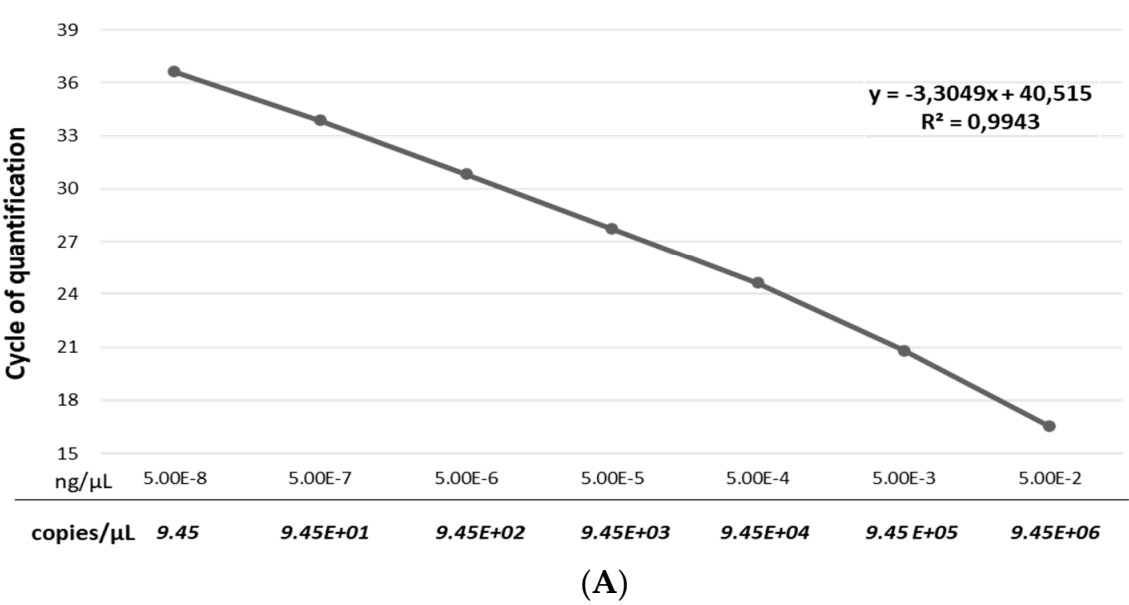

**(A)**

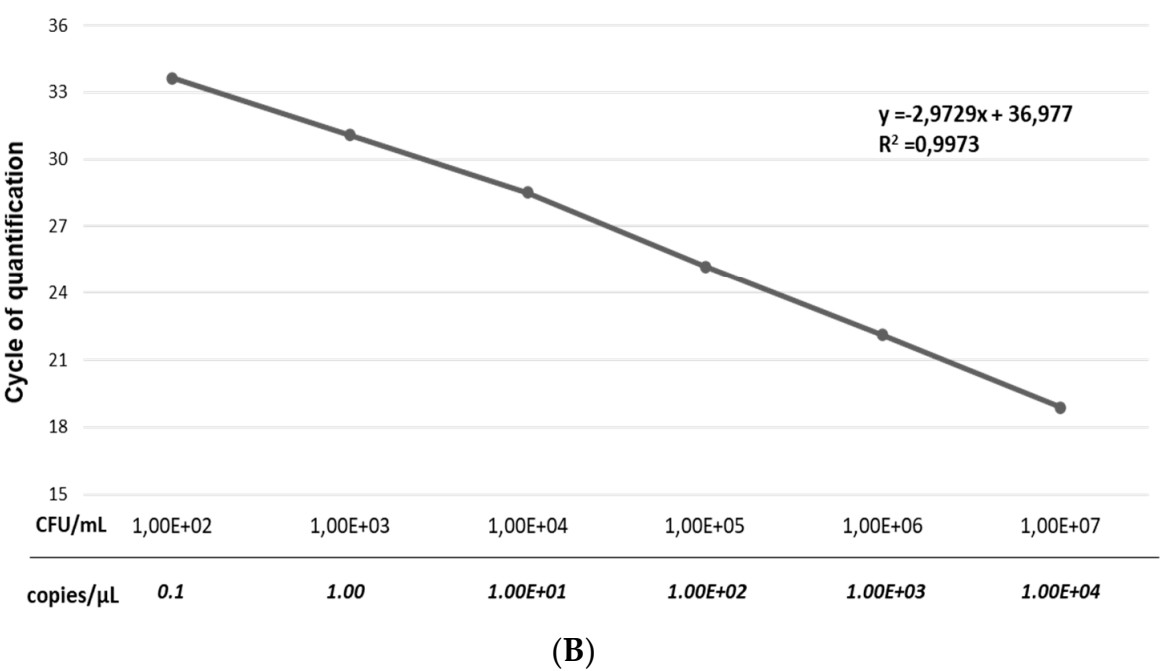

**(B)**

**Figure 2.** Standard curves generated by qPCR assay on *rimM* plasmid DNA (**A**) and bacterial suspension (**B**). Plasmid DNA was ten-fold serially diluted from 50 pg/μL to 50 ag/μL ($9.45 \times 10^6$–9.45 copies/μL). The bacterial suspension was ten-fold serially diluted from $1.00 \times 10^7$ to $1.00 \times 10^1$ CFU/mL ($1.00 \times 10^4$–0.1 copies/μL). On the y-axis: the cycle of quantification. (**A**) On the x-axis, ng/μL and copies/μL (italic and bold) are indicated. (**B**) On the x-axis, CFU/mL and copies/μL (italic and bold) are reported.

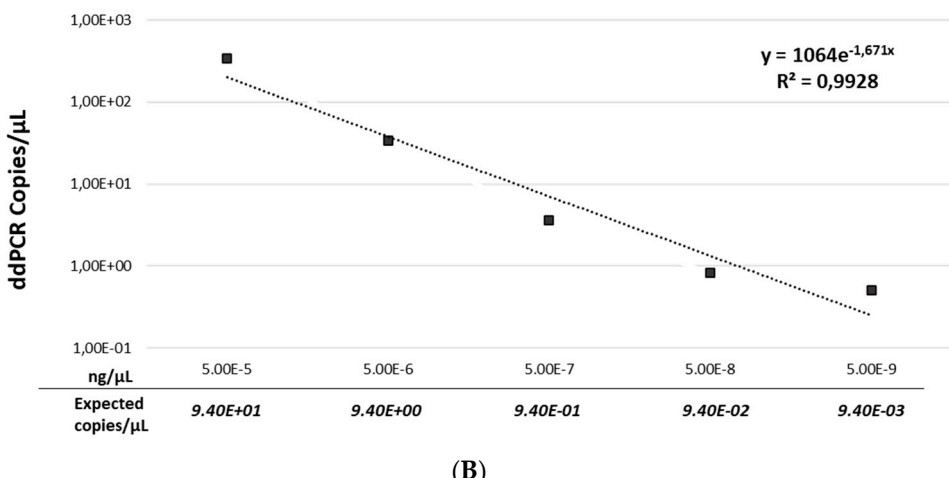

**Figure 3.** Linear regression of the ddPCR assays generated using the same 10-fold dilution series of bacterial suspension (**A**) and *rimM* plasmid DNA (**B**) tested with the qPCR assay. Number of copies/µL measured by ddPCR on the y-axis are correlated to (**A**) dilutions of bacterial suspension ranging from $1.00 \times 10^6$ to $1.00 \times 10^1$ CFU/mL, and (**B**) dilutions of plasmid ranging from 50 fg/µL to 5 ag/µL. Both figures indicated the corresponding expected copies/µL on the x-axis.

*3.5. Evaluation of the ddPCR Assay in Field Samples*

In this study, a comparison was carried out between the diagnostic performances of ddPCR and qPCR assays on naturally infected olive and insect samples. The DNA recovered from 34 olive and 27 insect samples was simultaneously tested by ddPCR and qPCR. Based on the Cq values generated in the qPCR assay, samples were clustered in three and two groups, respectively, for olives and insects. The olive samples were grouped as follows: (i) Cq from 27 to 29; (ii) Cq from 30 to 32; and (iii) Cq from 33 to 34. On the other hand, the insect samples were subdivided as follows: (i) Cq from 23 to 32 and (ii) Cq from 33 to 36. As shown in Table 5, all the olive samples belonging to the first and the second group were correctly identified as positive by ddPCR (19/19); while 13 out of 15 samples belonging to the third group (Cq 33–34), which had been classified as undermined by qPCR, were identified as positive by ddPCR. All insect samples included in the first group (11 specimens), yielding positive qPCR reactions, were also identified as positive by ddPCR (Table 6). Eight of the thirteen samples of the second group, classified as undetermined by qPCR, tested positive with ddPCR, two remained as undetermined samples, and three

tested negative (Table 6). Moreover, all insects and olive samples collected in *Xf*-free areas, used as NICs, tested negative with both assays. NTCs included in all runs gave no amplification and no fluorescence signal. Therefore, the accuracy level reached by ddPCR was 100% in comparison with that of qPCR, with an increment level of approx. 38 and 30% to detect *Xfp* in olive and insect samples, respectively.

**Table 5.** Number of naturally infected olive samples assessed by qPCR assay as positive and undetermined (grouped in different ranges of the Cq values) compared with number of positive and negative samples detected by ddPCR assay.

| Olive Samples Grouped Based on the Cq Values Obtained by qPCR | | | | ddPCR | | |
|---|---|---|---|---|---|---|
| Group | Cq Values | N. of Samples | qPCR Result | Positive | Negative | Total |
| (i) | 27–29 | N. 10 | Positive | 10 | 0 | 10 |
| (ii) | 30–32 | N. 9 | Positive | 9 | 0 | 9 |
| (iii) | 33–34 | N. 15 | Undetermined | 13 | 2 | 15 |
| | | | Total | 32 | 2 | 34 |

**Table 6.** Performance of ddPCR and qPCR assays for detection of *Xfp* in naturally infected insects. Insect samples were divided into two groups according to the Cq value obtained by qPCR.

| Insect Samples Grouped Based on the Cq Values Obtained by qPCR | | | | ddPCR | | | |
|---|---|---|---|---|---|---|---|
| Group | Cq Values | N. of Samples | qPCR Result | Positive | Negative | Undetermined | Total |
| (i) | 23–32 | N. 11 | Positive. | 11 | 0 | 0 | 11 |
| (ii) | 33–36 | N. 13 | Undetermined | 8 | 3 | 2 | 13 |
| (iii) | >36 | N. 3 | Negative | 0 | 3 | 0 | 3 |
| | | | Total | 19 | 6 | 2 | 27 |

## 4. Discussion

Digital droplet PCR is one of the newest PCR formats, part of the third generation of the PCR techniques, and is currently becoming widely used in different applications requiring detection and/or quantification of the target nucleic acid, as well as to check for gene mutations and DNA modifications [26], to analyze gene expression [27], and to detect and quantify human and animal parasites and pathogens [28–35]. Digital droplet PCR technology enables more precise quantification even at low template concentrations, not requiring a standard reference curve as the qPCR does. The large-scale partitioning of the template in ddPCR increases the precision of quantification and reduces interference due to PCR inhibitors. Several studies have reported its implementation in plant pathology [36] for the quantitative detection of plant fungi [37–39], bacteria and fastidious bacteria [24,40–45], phytoplasma [46], viruses [47,48], and viroids [49]. For *Xf*, a ddPCR protocol was developed by Dupas et al. (2019) [24]. This assay was developed using strains of the subspecies multiplex, with olive and various ornamental plants as host species. In our work, we implemented ddPCR into testing insect vectors; several crop species; and oleander, one of the host plants with a high content of endogenous phenolic compounds whose oxidation generates PCR inhibitors during plant sap preparations. In this study, qPCR and ddPCR assays based on worldwide primer sequences validated [11] and targeted the highly conserved region of the *rimM* gene. These tests were used for the detection of one of the most aggressive bacterial genotypes (*Xfp* ST53), and ddPCR was validated for the first time in the detection of this bacterium in insect vectors and important crop species (almond, citrus, and grapevine). To estimate the performance, the LOD and LOQ of the protocols herein improved sufficiently to detect *Xfp* in six matrices; a panel of artificially contaminated samples with the DNA of a *Xf*-recombinant plasmid or an *Xfp* bacterial suspension wase used. The best diagnostic performance (high fluorescence amplitude and a better separation

between positive and negative droplets) of the ddPCR was achieved by modifying the reaction conditions previously reported by Dupas et al. (2019) [24]. Optimized conditions included lower primer concentrations (600 nM), higher probe concentrations, lower DNA templates (4 µL for plants and 6 µL for insects), and an increased number of cycles—up to 45 (as suggested by Huggett et al. (2013) and Whale et al. (2020) [20,21]). As shown in Supplementary Figure S1, the use of low volumes (2 µL) of plant/insect extracts to set up the ddPCR reactions generated poor results, especially when testing samples with low levels of pathogen contamination, i.e., samples yielding Cq values ranging from 32 to 34 in qPCR. On the other hand, the use of a high volume of extracts (8 µL) did not improve either the amplitude value or the droplet pattern, but it nevertheless generated a 'droplet-rain', which caused difficulties in the interpretation of the results. Moreover, the modifications introduced to the qPCR official test (PM 7/24 (4)—Annex V [15]) (i.e., half the reaction volume, avoidance of BSA) proved not to impair the efficiency or analytical sensitivity of this qPCR assay while also reducing the reagents' input requirements. All standard curves generated by qPCR assay had coefficients of determination ($R^2$) above 0.99% and efficiency values within the optimal range of 90–120%, indicating the high performance of the assay. The efficiency for the matrix *N. oleander*, nearly reaching its limit of 120%, is probably due to the presence of PCR inhibitors and contaminants in this matrix. Furthermore, the efficiency of the standard curve generated by qPCR for the olive matrix in this work was strongly improved as compared to that obtained by Dupas et al. (2019) (105.38% vs. 238.14%). The analytical sensitivity of the qPCR test was fixed at 50 ag/µL, corresponding to 9.45 copies/µL; and 100 CFU/mL, corresponding to 0.1 copies/µL for plasmid DNA and bacterial cells, respectively. The detection limits of the ddPCR were determined to be $4.30 \times 10^{-1}$ copies/µL and $5.06 \times 10^{-1}$ copies/µL for bacterial suspensions and plasmid DNA, respectively. A high correlation between the two molecular techniques was observed (r = 1). Overall, ddPCR showed higher analytical sensitivity than qPCR, since it detected *Xfp* at a lower dilution than qPCR for *O. europea*, *C. sinensis,* and *N. oleander*. ddPCR, for the insect matrix, showed a LOQ of 10 CFU/mL (corresponding to $9.22 \times 10^{-1}$ copies/µL) and a LOD of $10^2$ CFU/mL (corresponding to 2.02 copies/µL), since variable results were obtained by the different replicates of the lower dilution. Both methods showed the same limit of detection only for *V. vinifera* and *P. dulcis*. A preliminary evaluation of the potential applications of the new, improved ddPCR assay was also carried out on 34 olives and 27 insects, all naturally infected. ddPCR clearly proved to be a powerful diagnostic tool to solve those cases in which other diagnostic approaches are unable to assess the *Xf* status. For example, in our tests, ddPCR was able to classify 13 olive and 8 insect samples as positive that were classified as undetermined by qPCR (Cq value = 33–34 for olive; Cq value = 33–36 for insect). Thus, improving the efficiency and the early detection of this bacterium represents the next challenging step for research programs. However, ddPCR technology requires very expensive reagents and equipment, is more time-consuming, and needs more attention paid to sample handling than qPCR. Thus, with the current technology, it is not suitable for routine analysis. However, as shown in our work, it can be very useful for supporting diagnostic responses in samples that yield qPCR results that are difficult to interpret (i.e., close to the threshold and the LOD), reducing the number of samples with inconclusive results.

## 5. Conclusions

To the best of our knowledge, this is the first report concerning the application of the ddPCR technique to detect *Xf* in insect vectors and in several important crops. The results gathered in this study demonstrated the potential and the high sensitivity of the ddPCR assay compared to the established qPCR assay, and hardly more than two droplets were needed to confirm a sample to be positive with ddPCR. Therefore, it is a reliable method for the absolute quantification of target DNA, especially at low levels of infection of the bacterium and in the presence of high levels of PCR-interfering compounds in plants and insects. Moreover, the ddPCR protocol implemented in this study proved to be suitable for

determining the sanitary statuses of samples yielding inconclusive results with qPCR. Thus, the use of this newly implemented test becomes relevant in supporting the assessment of critical samples in the official controls, i.e., samples from plants at an early stage of infection whose results by qPCR are inconclusive.

**Supplementary Materials:** The following supporting information can be downloaded at: https://www.mdpi.com/article/10.3390/agriculture13030716/s1, Figure S1: Assessment of optimal olive and insect DNA amount in ddPCR reaction mix; Figure S2: Calibration curves of qPCR; Figure S3: Linear regression curves of the ddPCR assay.

**Author Contributions:** Conceptualization, methodology, data curation, software, writing—original draft preparation: S.S.A. and G.L.; samples and insect resources: A.L., V.C. and S.S.A.; writing—review and editing: G.L., S.S.A., M.S. and A.M.; supervision, G.L., M.S. and A.M.; project administration and funding acquisition, G.L., A.M., M.S. and D.B. All authors have read and agreed to the published version of the manuscript.

**Funding:** This research was carried out in the framework of the Project BEXYL (Beyond Xylella, Integrated Management Strategies for Mitigating Xylella fastidiosa impact in Europe; grant number 101060593) funded by the European Union's Horizon Europe research and innovation program. This work has received funding from the EC through the H2020-FET-OPEN-2018–2019-2020-01 Project FREE@POC (Towards an instrument-FREE future of molecular diagnostics at the Point-Of-Care'), grant number 862840.

**Institutional Review Board Statement:** Not applicable.

**Informed Consent Statement:** Not applicable.

**Data Availability Statement:** Not applicable.

**Conflicts of Interest:** The authors declare no conflict of interest. The funders had no role in the design of the study; in the collection, analyses, or interpretation of data; in the writing of the manuscript; or in the decision to publish the results.

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
