# Peer review of "Detection of Xylella fastidiosa in Host Plants and Insect Vectors by Droplet Digital PCR"

_agriculture, doi:10.3390/agriculture13030716_

Round 1
Reviewer 1 Report
Unfortunately the presented manuscript is severely flawed and therefore I cannot recommend it for publication. Whilst the work presented is sound and the experiments adequate, extensive further laboratory work is required to enable the authors to include all of the data that is required for this type of publication. The presented paper does not allow the reader to assess the suitability of the presented diagnostic test for use.
The widely used Harper et al., X. fastidiosa qPCR assay has already been converted to the ddPCR format and reported within the literature by Dupas et al., 2019. In this study the authors state that they have ‘optimised’ and ‘validated’ the assay. Unfortunately, no data is presented to allow the reader to assess either the optimisation or validation of the assay in the newly presented format. The data presented is only of the authors modification and there is no comparative testing with the first reported ddPCR. Indeed even the included comparison with the qPCR assay is flawed as the authors say this was used in a ‘slightly modified’ format. The reader is left having to trust that the amended ddPCR protocol is an improvement to the Dupas paper, they cannot assess themselves.
The authors have altered two of the most fundamentally important parameters of any molecular diagnostic assay – the number of cycles and the concentration of primers and probe. Any alteration to the parameters of an assay such as these require the complete revalidation of the assay, as the performance characteristics may have changed. The authors have solely focused on the sensitivity of the assay. However the sensitivity will be irrelevant if the changes made mean that the specificity of the assay has been altered. The authors must evaluate the changed/modified assay as if they were developing a new assay – they must test as a minimum, specificity, inclusivity and sensitivity. One without the others is useless. The authors have not completed any inclusivity testing – the entire manuscript uses a single subspecies of X. fastidiosa subsp. pauca. This is not acceptable unless the manuscript title is changed so that the assay is described as only being for X. fastidiosa subsp. pauca and not X. fastidiosa generically.
Furthermore, despite the extensive sensitivity testing using artificially spiked matrices the authors have not done any sensitivity testing on naturally infected samples, despite the fact they have access to these. Undertaking 10-fold dilution series on the naturally infected plant and insect samples (with the dilutions prepared in a background of healthy plant/insect) and testing these with both qPCR and ddPCR would be highly informative. Whilst testing with artificially generated samples is standard practice, especially when naturally infected samples are not available, this type of test cannot replicate naturally infected samples, and where these are available, they should always be used in parallel to ensure the robustness of the data.
It is clear from the author’s references to EPPO PM 7/24 that they are aware of the requirements for development and validation of methods suitable for use in plant pathogen quarantine outbreaks. The abstract even says that they have extended the validation of the assay. Firstly, they cannot be ‘extending the validation’ when the have changed the assay - this is no longer the same assay as that in the Dupas paper. Secondly, as discussed above, there is a complete lack of validation presented, sensitivity data alone does not mean an assay is validated.
Furthermore, the authors abstract implies much more comprehensive work has been undertaken. I is implied that this is an improvement from Dupas et al., 2019 however these is no data presented to allow the reader to determine this. I struggle to see how this manuscript adds much to the field of Xylella diagnostics, given that the method presented will not be able to be used by any diagnostic lab without extensive validation.
Reviewer 2 Report
The reviewed manuscript is dedicated to the design and validation of a ddPCR-based assay detecting Xylella fastidiosa. Currently, sensitive and relatively simple qPCR is the gold standard for diagnostics of that pathogen. However, more sensitive methods are required for more precise diagnosis. Here, authors improved the conditions of the previously reported ddPCR assay
And validated their specificity and sensitivity on both artificial and real samples models. The manuscript is well-written and the presented results are timely and interesting for scientists, specializing on the field of molecular diagnostics. However, a few comments need to be made and addressed.
Major issues:
1. In the introduction, authors are encouraged to provide more arguments for the usage of more sensitive ddPCR instead of more economical qPCR.
2. Authors are encouraged to provide more information about a previously developed ddPCR assay (10.1016/j.mimet.2019.05.010) and underline the reasons for its optimization.
Minor issues:
1. Page 3 lines 118-119 “Negative and positive internal controls (NIC, PIC) were prepared and processed for each matrix.” — authors are encouraged to describe how the controls were prepared.
2. Matrix, matrices — normally, the term “matrix” designates a substance in a space between eukaryotic cells. Therefore, its usage with another meaning would be confusing. Possibly, another term would improve the readability of the manuscript.
3. Page 4, line 147: “using primers/probe by Harper et al” — were the primers tested on the presence of any SNP?
4. Page 5, line 203: “4 and 6 µL of DNA” — what was DNA amount in ng or copies?
5. Page 6, line 228: “1.00E+02 CFU/mL, corresponding to 0.1 copies/µL” — is it a concentration in an analyzed sample or in the qPCR reaction?
6. Page 8, lines 273-274: “V. vinifera and P. dulcis” — here and in some other places species names are need to be italicized.
Reviewer 3 Report
The manuscript reports important findings on the application of droplet digital PCR to detect Xylella fastidiosa (Xf) in various host plants and insect vectors. Overall, the manuscript is well-written and easy to follow. The experiment section is described in detail and the results are clearly explained.
The detection of and quantification of Xf using ddPCR had been published previously. In the current study, the authors optimized and validated the use of ddPCR for the detection and quantification of Xf in different susceptible plant species and insect vectors. Relevant, but not quite interesting since there is not much novelty here. The application ddPCR for the detection and quantification of Xf in more plant species and insect vectors. The paper is well written and the text is clear and easy to read. The conclusions are consistent with the evidence and arguments presented. And the authors addressed the main question posed.
